# Mechanical Performance of Chairside Ceramic CAD/CAM Restorations and Zirconia Abutments with Different Internal Implant Connections: In Vitro Study and Finite Element Analysis

**DOI:** 10.3390/ma14175009

**Published:** 2021-09-02

**Authors:** Sergio Giner, José F. Bartolomé, Pablo Gomez-Cogolludo, Carlos Castellote, Guillermo Pradíes

**Affiliations:** 1Facultad de Odontología, Universidad Complutense de Madrid, 28660 Madrid, Spain; sergioginergarrido@hotmail.com (S.G.); pablogom@odon.ucm.es (P.G.-C.); gpradies@odon.ucm.es (G.P.); 2Instituto de Ciencia de Materiales de Madrid, Consejo Superior de Investigaciones Científicas, 28660 Madrid, Spain; 3Escuela de Arquitectura, Ingeniería y Diseño, Universidad Europea de Madrid, 28660 Madrid, Spain; carlos.castellote@universidadeuropea.es

**Keywords:** fatigue, dental implant, biomechanics, finite element analysis

## Abstract

(computer-aided design-computer-aided manufacturing) CAD/CAM monolithic restorations connected to zirconia abutments manufactured with a chairside workflow are becoming a more common restorative option. However, their mechanical performance is still uncertain. The aim of this study was to evaluate the mechanical behavior of a combination of a zirconia abutment and monolithic all-ceramic zirconia and lithium disilicate crown manufactured with a chairside workflow, connected to titanium implants with two types of internal connection—tube in tube connection and conical connection with platform switching. They were thermally cycled from 5 °C to 55 °C and were subjected to a static and fatigue test following ISO 14801. The fractured specimens from the fatigue test were examined by SEM (scanning electron microscopy). Simulations of the stress distribution over the different parts of the restorative complex during the mechanical tests were evaluated by means of (finite element analysis) FEA. The mechanical performance of the zirconia abutment with an internal conical connection was higher than that of the tube in tube connection. Additionally, the use of disilicate or zirconia all-ceramic chairside CAD/CAM monolithic restorations has similar results in terms of mechanical fracture and fatigue resistance. Stress distribution affects the implant/restoration complex depending on the connection design. Zirconia abutments and monolithic restorations seem to be highly reliable in terms of mechanical resistance.

## 1. Introduction

Nowadays, osseointegrated implants are becoming the most common treatment alternative among clinicians in order to restore missing teeth as they have been proven to have high survival rates in the scientific literature. Conventionally, these implants were restored using titanium abutments and porcelain fused to metal crowns. However, using a metallic component has numerous drawbacks. The aesthetic result of the restoration could be dimmed by the absence of naturalness, and not only the white aesthetics could be jeopardized, but the pink aesthetics could also be affected if a grey shadow appears in the gingival margin. The use of materials with improved biomimetic characteristics such as dental ceramics has allowed us to achieve better aesthetic results regarding the appearance of our restorations [1,2,3,4,5,6,7,8].

Developments in ceramic materials, equipment, and technology have made it possible to manufacture an indirect aesthetic restoration in a single appointment. CAD/CAM was created to give the clinician the possibility of designing and manufacturing a restoration without the need for laboratory assistance, provisional restorations, or conventional impressions [9]. The computer assisted ceramic reconstruction (CEREC) system introduced by Sirona Dental Systems, Inc., known today as Dentsply Sirona, was the first chairside operational system available thanks to the work of Mörmann and Brandestini [10]. The survival rates and clinical success of crowns manufactured using a chairside workflow has been demonstrated. Different studies report between 83.5% and 95.5% survival rates at ten and nine years [11,12]. The CAD/CAM blocks are manufactured in optimal conditions, creating a restoration with greater intrinsic resistance and reliability that can be found in laboratory-manufactured restorations with conventional materials [13,14]. Rosentritt et al. [14] studied the behavior of monolithic crowns made in different materials when a chairside workflow or a conventional workflow are followed. These crowns were cemented on a titanium abutment. The results obtained for both groups were similar, with no significant differences. However, there are few studies that use ceramic abutments and restorations manufactured with a chairside workflow, so there is a lack of information on their behavior and reliability.

The implant connection should also be considered as is an important variable of the restorative complex; traditionally, hexagonal external connections were used. However, in recent years, internal connections have become popular as they preserve more crestal bone. In spite of this, there is a wide range of internal connections, and their biological and mechanical properties are not the same. Conical connections have gained the upper hand in recent years due to their internal fit and less screw loosening. These conical connections incorporate the concept of platform switching. This consists of the placement of abutments or other prosthetic elements with a narrower diameter than that of the implant platform. In the last decade, many clinical and histomorphometric studies have been published to demonstrate the efficacy of this procedure in preserving peri-implant bone [15,16,17,18,19,20]. All conclude that there is less bone loss in those cases in which it has been processed with platform switching compared with the cases that are resolved with the conventional platform matching model. However, the use of a narrower abutment could compromise the mechanical behavior of the complex [21,22,23].

Therefore, zirconia abutments with a titanium base are promising candidates to substitute for titanium abutments from an aesthetic and biological point of view. However, information on the mechanical performance of zirconia abutments supporting ceramic crowns and the influence of the implant connection type is limited. Because of this scarcity of mechanical in-vitro studies and lack of significant clinical studies, the mechanical performance of chairside CAD/CAM zirconia implant abutments supporting translucent lithium disilicate and zirconia monolithic crowns should be analyzed. The aim of this study is to evaluate the mechanical behavior of chairside customized zirconia abutments with two different implant connections under static and fatigue conditions. Moreover, in vitro tests were reproduced using engineering tools like von Mises and finite element analysis, as they can consider all the different variables (type of load and material) and calculate the stress distribution of a system in which dental implant components are involved [24,25].

The first null hypothesis is that no differences in mechanical performance would be seen between the systems with the two types of connection (tube in tube and conical connection). The second null hypothesis is that no differences in mechanical behavior would be registered between the systems with the two different crown materials (lithium disilicate and zirconia).

## 2. Materials and Methods

This in vitro study (not involving any human or animal participation) included 30 Camlog Screw Line implants on a Promote plus surface (4.3 mm diameter and 11 mm height) with a transmucosal part of 0.4 mm, with a tube in tube trilobular connection, and 30 Conelog Screw Line implants (4.3 mm diameter and 11 mm in height) with a conical connection. For both implants, a customized zirconia abutment with a titanium base was used. Two subgroups (*n* = 15) were included depending on the material of the crown: group zirconia (Group Z) and group lithium disilicate (Group LD):Subgroup Tube ZZ: 15 specimens, using a tube in tube connection implant and a customized zirconia abutment with a monolithic zirconia CAD/CAM crown with standardized anatomy of a maxillary premolar.Subgroup Tube ZLD: 15 specimens, using a tube in tube connection implant and a customized zirconia abutment with a monolithic lithium disilicate CAD/CAM crown with standardized anatomy of a maxillary premolar.Subgroup Conical ZZ: 15 specimens, using a conical connection implant and a customized zirconia abutment with a monolithic zirconia CAD/CAM crown with standardized anatomy of a maxillary premolar.Subgroup Conical ZLD: 15 specimens, using a conical connection implant and a customized zirconia abutment with a monolithic lithium disilicate CAD/CAM crown with standardized anatomy of a maxillary premolar.

### 2.1. Specimens Manufacture

For the manufacture of the restorations, two replicated models; one with a replica of Camlog Screw Line implants Promote plus surface (diameter 4.3 mm in position 1.4) and the other with a replica of Conelog Screw Line implants (diameter 4.3 mm in the same position) were used.

For the digitization of the casts, the digitalization unit Cerec Onmicam Intraoral Scan (Sirona) was used. A prior wax up of the restoration was performed with the aim of standardizing the anatomy of all of the crowns. After the digitalization of both models, we proceeded to design the restorations choosing the biogeneric copy option with the software CEREC 4.4 (Dentsply Sirona, York, PA, USA). A special scanbody (Dentsply Sirona, York, PA, USA) was used to locate the replicas directly on the titanium bases for the zirconia abutments. Once the models were digitalized, we proceeded to design an anatomical zirconia abutment for the Camlog Titanium base CAD/CAM and for Conelog Titanium base CAD/CAM with the same shape and anatomy.

Once the abutments were designed, they were milled using the MCXL milling unit of the Cerec system. Pre-sintered blocks of Incoris Zi (Dentsply Sirona), as well as the following milling burs were used: Step Bur 20 and Cylinder Pointed Bur 20. For complete sintering of the zirconia, the In-Fire Htc Speed (Dentsply Sirona, York, PA, USA) oven, which is specific for this material, was used in order to obtain a correct shrinkage of the material. All-ceramic monolithic crowns were milled from pre-sintered blocks of Emax-CAD (Ivoclar Vivadent, Schaan, Liechtenstein) lithium disilicate and Incoris Tzi (Sirona) zircona. For the crystallization of the lithium disilicate, the Programat CS2 (Ivoclar Vivadent, Schaan, Liechtenstein) oven was used. For the complete sintering of the zirconia, the In Fire Htc Speed (Sirona) oven, which is specific for this material, was used in order to obtain the correct shrinkage of the material.

Before cementation, the internal and external surfaces of the zirconia abutments and the internal surfaces of the zirconia crowns were sand-blasted with a sandblasting unit for clinical use (Microetcher II^®^, Danville Materials) with 50 microns aluminium oxide particles at a pressure of 1 atm, and they were cleaned in an ultrasonic cleaning unit. The lithium disilicate crowns were treated with 4.5% hydrofluoric acid for 20 s, washed up with water and cleaned in an ultrasonic unit, then silanized with monobond Plus (Ivoclar Vivadent). The zirconia abutments were luted to the titanium CAD/CAM base with a dual polymerization resin cement speed Cem Plus (Ivoclar Vivadent), positioning it with hand pressure. Before luting the crowns, a randomization table was performed in which the implant, abutment, and crown were involved. Once the randomization was completed, each abutment was screwed to the corresponding implant with an insertion torque of 20 N·cm, following the manufacturer’s recommendations, using a torque wrench and sealing the screw access with polytetrafluoroethylene (PTFE).

Dual polymerization resin (Speed Cem Plus, Ivoclar Vivadent) was used to cement the crowns. To standardize the cementation phase, all restorations were cemented using a clamping jaw and a torque wrench with a pressure of 40 N/cm^2^. To successfully remove the excess cement, a photopolymerization lamp of 650 Mw/cm^2^ (Bluephase S, Ivoclar Vivadent) was used for a time between 2 and 4 s for each surface of the crowns and the abutment. Once the excess was removed, the photopolymerization was completed by 40 s per surface. Representative pictures for each group are presented in Figure 1.

### 2.2. Mechanical Test

The aging of the specimens was carried out using an automated thermocycling device for dental materials [26]. The specimens were subjected to a total of 10,000 cycles at a temperature of 5 ± 5° and 55 ± 5° with an immersion time of 20 s in an artificial saliva solution [27].

All of the specimens were mounted perpendicular to the horizontal plane in the center of epoxy resin holder (EpoxiCure 2; Buehler, IL, USA) with a Young’s modulus Conelog Screw Line implants comparable to that of the human mandible [28], following the method previously described [29,30]. The specimens were introduced into the epoxy resin using positioners. These positioners, placed horizontally on the mold, allowed for leaving 3 mm of the implant exposed from the platform to the epoxy resin, simulating peri-implant bone loss according to the specifications of ISO 14801 [31].

The static tests were performed with a mechanical testing machine (AG-X Series, Shimadzu, Kyoto, Japan). The load was applied to the specimens’ occlusal surface of the palatal cup at a crosshead speed of 0.5 mm/min with an angle of 30° (Figure 2). Load (Newtons) and displacement (millimeters) data were collected during the test by a specific program (Trapezium X Software, Shimadzu, https://www.shimadzu.eu/trapezium-x-software, accessed on 18 May 2021). The test machine was programmed to stop the force test process for a greater displacement of 1.5 mm. The maximum force (F_m_) was considered as the load–bearing capacity (Figure 3). The force (F_p_) at which the load–displacement curve first deviated by 10% from the regression line in the elastic interval was recorded as an indicator for the initiation of plastic deformation (Figure 3) [29,32].

The mechanical fatigue tests were carried out using an electromagnetic testing machine (EMT-1kn-30, Shimadzu, Kyoto, Japan) in dry conditions operated under load control and with a sinusoidal wave form at 2 Hz. The load ratio (minimum to maximum loading ratio) was equal to 10 with a unidirectional and tilted (30°) pulsating load. A fatigue life test was carried out until the specimens showed failure, or it was interrupted at N_f_ = 2 × 10^6^ cycles (∼8 years of simulated function in the human masticatory system [33]) for the specimens without failure. The cyclic forces selected for the test were between − 40 and − 400 N, simulating forces generated in the oral cavity [34]. Six samples per composition were evaluated. The number of cycles to fracture was registered using a specific software program (Trapezium X^®^ software, Shimadzu^®^, Kyoto, Japan, https://www.shimadzu.eu/trapezium-x-software, accessed on 18 May 2021).

### 2.3. Statistical Analysis

Based on previous studies [29,30,32], the required sample size was calculated for a power of 80%, with an alpha value of 0.05, using specific software (G-Power version 3.1.9 for Mac OS, Heinrich Heine University, Dusseldorf, Germany).

Statistical analysis of all of the variables was carried out using SPSS^®^ software (version 25.0, SPSS Inc., Chicago, IL, USA). Descriptive statistics are expressed as means and standard deviations (SD) for the quantitative variables. A comparative analysis was performed by comparing fracture resistance and plastic deformation data among the four groups using Analysis of Variance (ANOVA). The statistical significance was set at *p* < 0.05.

### 2.4. Macro and Micro Structural Analysis

All of the tested specimens were photographed with a Canon 700 D with a 100 mm macro lens with extensor rings and an annular flash following a photographic protocol.

For the microstructural analysis of the deformed specimens after the static mechanical test, the samples of each group were embedded in a transparent resin (Tecmicro S.A., Madrid, Spain). The samples were cut on the longitudinal axis using a cutting machine with a diamond saw (Micromet M, Remet, Italy). The final slice of each specimen was polished with silicon carbide papers and finally with a silica suspension.

The microstructural study of the deformed specimens after the static mechanical test and the fatigue fracture surfaces of the broken samples were examined using SEM (Phenom G2 pro, Eindhoven, Holland) at different magnifications.

### 2.5. Finite Element Analysis

A finite element analysis was conducted in which different simulations were carried out for the different restorative options. It was intended to obtain the values in terms of the stress for each of the specimens, and to compare them with each other in static and cyclic fatigue conditions according to the ISO 14801. The CAD models of each piece were modelled in Solidworks 2019 (Dassault Systèmes, Waltham, MA, USA) and were imported into Ansys Workbench 2019 (ANSYS, Inc., Canonsburg, PA, USA), where the corresponding finite element models were created, and different properties were applied to each piece depending on the material, boundary conditions, contacts, existing loads, stereolithography (STL) mesh, etc. The element was defined by 10 nodes with three degrees of freedom at each node: translations in the nodal x, y, and z directions. The element had plasticity, hyperelasticity, creep, stress stiffening, large deflection, and large strain capabilities. It also had a mixed formulation capability for simulating deformations of nearly incompressible elastoplastic materials, and fully incompressible hyperelastic materials. The material properties were assumed to be homogeneous, isotropic, and linearly elastic, and the applied state equation was the Hooke Law through Young and Poisson Modulus specification (Table 1). The total number of nodes, elements, and minimum edge length are described in Table 2.

The crowns that were cemented to both the abutment or Ti base, as well as the implant to the mold, are considered to have a “bonded” type contact. For the rest of the pieces, a “friction” type contact was considered between the materials. It was verified that there were no contact opening problems. The three contact surfaces between the testing machine and the epoxy resin were simulated as a fixed support. The preload (first torque) of the screw was previously adjusted in order to ensure the reality in the application and the distribution of the Von misses forces, and then the external load was applied according to the ISO norm. Therefore, the state of the contacts and the accuracy of the created mesh were analyzed before considering the created models to be correct. Finally, von Mises stress maps and safety factor maps were performed in FEA.

## 3. Results

### 3.1. Static Load

The Conical ZZ subgroup obtained a maximum load (F_m_) of 1050 ± 92 N and the Conical ZLD subgroup obtained a F_m_ = 1102 ± 112 N. The Tube ZZ subgroup obtained a F_m_ = 1015 ± 73 N and the Tube ZLD subgroup obtained a F_m_ = 975 ± 64 N. There were no statistical differences (*p* > 0.05) between the subgroups. However, the highest values were registered in the conical connection group. No abutment or crown was fractured during the static test. On the other hand, significant differences (*p* < 0.05) were observed in the load at plastic deformation (F_p_) among the different connection configurations (Tube ZZ = 552 ± 87 N, and ZLD = 563 ± 66 N vs. Conical ZZ = 704 ± 44 N and ZLD = 682 ± 64 N). The highest values were registered in the conical system.

### 3.2. Fatigue Behavior

For the tube group during the fatigue test, we noticed that no specimen was capable of supporting 2 × 10^6^ cycles at a maximum load of 400 N (Table 3). The Tube ZZ subgroup survived a mean ± standard deviation of 403.673 ± 85.398 cycles at 400 N and the Tube ZLD subgroup survived a mean ± standard deviation of 254.577 ± 57.789 cycles. No abutments or crowns were fractured during the fatigue test.

For the conical groups, neither the ZZ subgroup, nor the ZLD subgroup failed during the fatigue test at 400 N, which means that the specimen had an infinite fatigue life at this load.

### 3.3. Failure Modes

During the static load test for all of the groups, no fracture of any of its components was observed, but there was deformation of the abutment and the prosthetic screw (Figure 4). The experiment was stopped when there was a 1.5 mm deformation, which was considered as the specimen having failed.

During the fatigue test, no specimen from the conical connection subgroups failed. Regarding the tube connection, all of the specimens showed the same type of failure: implant at the connection level at one of the indexing points where the walls were thinner and the screw fracture. Figure 5 and Figure 6 show a representative fractured specimen of the tube connection groups after the fatigue tests. None of the abutments or crowns were fractured. The weakest point seemed to be the abutment screw that absorbed all the applied tension until fracture. After that, the system flexed, producing an implant neck fracture (Figure 5). Figure 6 shows an SEM micrograph of the fracture surface of the prosthetic screw. Two well-defined regions were found: In Figure 6a, the region characterized by lines of slender grooves perpendicular to the load direction (Figure 6c), and in Figure 6b, the region with a dimple pattern corresponding to the catastrophic ductile failure (Figure 6d).

### 3.4. Finite Element Analysis

The von Mises stress indicated how the load was distributed on the different surfaces and, consequently, which areas were more susceptible to stress. A quantitative view allows for perceiving an increase in stress in the tube connection subgroups when compared with the conical connection subgroups when a static load of 900 N is applied (Figure 7). It can be seen that the tube connection model (Figure 7a) appeared to achieve higher von Mises values than the conical connection model (Figure 7b). For the tube connection models, the stress distribution was more homogeneous. This effect was particularly relevant at the screw level (Figure 8). At a cyclic load of 400 N, the screw suffered a maximum tensile stress of 146 MPa in the conical connection model and 462 MPa in the tube connection model. In the case of conical connection subgroups, the forces were more distributed on a wider surface, and the internal screw seemed less affected by the mechanical stresses. The fatigue safety factor map at 2 × 10^6^ cycles is shown in Figure 9. In the tube connection simulated models, it was observed that the screw was more susceptible to the fatigue component, and therefore was the most critical. The screw was the piece that first broke and the one that therefore limited the life of the whole assembly. Consequently, the fatigue life of the conical connection models was much higher than the tube connection models. The material of the crown did not show a relevant difference. Therefore, the FEA stress models are consistent with the results of the in vitro mechanical tests.

## 4. Discussion

Statistically significant mechanical behavior differences have been reported between the systems with the two types of connection (tube in tube and conical connection). On the other hand, no differences in mechanical behavior were registered between the systems with the two different ceramic restoration materials (zirconia and lithium disilicate). Therefore, the first null hypothesis was rejected and the second null hypothesis was accepted.

A challenging aspect in treatments with dental implants is the placement and subsequent restoration in high aesthetics areas, mainly due to the fact that the level of peri-implant bone support and soft tissue dimensions are critical factors in the final result. It goes without saying that from a biological and histological point of view, the platform switching is, without a doubt, indicated. However, the effect of placing narrower CAD/CAM zirconia implant abutments with conical implant−abutment connections for platform switching supporting single all-ceramic crowns made with a chairside workflow on the transmission of forces along the implant−abutment complex and its corresponding mechanical behavior is unknown. In this research, one of the study variables was the implant design. Two types of configurations have been studied. On the one hand, a cylindrical connection with a regular platform and without platform switching using an abutment of the same diameter, on the other hand, a conical and the concept of platform switching incorporated into its design, forcing the use of a narrow abutment for the restoration. Logic would suggest that placing a narrower abutment of a smaller diameter would increase the concentration of the stresses at the level of the abutment neck and affect its mechanical performance. However, the results of this investigation indicate the opposite.

In the case of the static test, the mean fracture resistance and plastic deformation point of the zirconia abutment with an internal conical connection group were higher than that of the zirconia abutment with the tube in tube connection. However, in both cases, these were still greater than the typical masticatory forces (148–354 N) [36,37], and therefore, both types of configurations seemed to present a very low risk of sustaining catastrophic fracture during their lifetime.

After performing the fatigue tests, a radically different behavior was observed between both implant configurations. While the study groups with tube implants largely failed during the tests, conical implants survived 2 × 10^6^ at a load as high as 400 N. As we have already mentioned, the weakest component of the whole restorative complex was the screw.

The fact that there was so much difference between both implants from a mechanical fatigue point of view lies in the fact that the transmission of forces to the clamping screw was higher in the tube implant than in the conical implant. This may be due to the difference existing in terms of the design of its internal connection, a cylinder versus a cone, although a narrower abutment was used. The failure pattern of the tube implants during the fatigue tests was the same in most specimens. With the microstructural analysis, it has been observed that, in addition to the implant, in all of the specimens, the screw was fractured. With the scanning electron microscope, it was possible to observe the areas already mentioned previously that showed the screw fatigue fracture. This suggests that the component that failed in the first place was the screw, perhaps fracturing before the implant. On the other hand, the tube connection and its indexing system with three edges on the implant platform mean that not only tensions were concentrated at this level, but also that the implant walls were thinner. The consequence, as it has been observed in the macro-structural analysis, was a vertical fracture that began at the level of these notches in a crown-apical direction.

Therefore, the sensitivity of tube connection groups to cyclic stresses was higher than for the conical groups. In a fatigue loading regime (in vivo mastication), a monolithic all-ceramic zirconia or lithium disilicate crown with a zirconia abutment manufactured with a chairside CAD/CAM connected to a titanium implant with an internal tube in tube connection may fail at a typical force generated in the oral cavity (≈400 N) in a very short period of time (≈400,000 cycles). This critical maximum load value is close to the average occlusal force applied at the cusp during chewing and swallowing for humans in normal conditions [38,39]. On the other hand, for the conical connection, system fatigue fracture did not occur in response to this stress level, and cycling at this maximum load would give infinite life. Moreover, the cyclic loading could also affect the formation of microgaps at the implant–abutment interface, resulting in large differences in the overall contact areas [40,41]. The oral microbiome can proliferate in this microgap and affect peri-implant tissues, causing inflammation and peri-implant diseases [42,43,44]. Several studies [45,46,47] have demonstrated that the internal conical implant–abutment connection is mechanically more stable than a flat-to-flat one, and is able to provide a better seal.

In this study, both the customized zirconia abutment and the crowns were manufactured in ceramic materials following a chairside workflow. In a study conducted by Joda et al. [48], in which a digital workflow was compared with a conventional workflow in implant prostheses, it concluded that the chair time was reduced, as the laboratory time that followed a digital workflow was significantly less. In addition to saving time, they concluded that with a sample of 40 reconstructions, that is, 20 per group, the digital workflow represented an environmental saving of 18% and that the major differences were in the cost of the laboratory.

It can be summarized that with a digital workflow, it is possible to optimize the treatment in order to achieve a reasonable cost−benefit ratio for the unit implant restorations. Keeping this in mind, it is more justified to follow a digital workflow. As the laboratory costs represent the greatest expense, using a chairside system to manufacture the restorations can greatly increase the benefit to the clinic. However, these systems require a learning curve that covers different phases, beginning with the impression taking using an intraoral unit, and continuing with the design of the restoration, which requires certain familiarity, and the handling of the software and the manufacture and sintering of the restoration for which some technical skills are needed. In addition, corresponding ovens for the correct sintering of the different materials are required. In this study, we decided to follow a chairside workflow using the CEREC system, which has its own intraoral, design, and manufacturing unit. Likewise, different materials have been used: zirconium oxide for the manufacture of a customized abutment, and translucent zirconia and disilicate for the manufacture of the crowns. In our study, the results are more than promising, leaving this work system in a very good position, despite all of the pertinent precautions having been taken regarding the design and manufacture of the restorations. Among the results obtained and assessing the type of catastrophic failure, it should be noted that neither the abutment nor crown fractured during any of the tests. It should be remembered that they were subjected to loads of up to 400 N.

The findings of this study suggest that the fracture resistance of the zirconia abutment with an internal conical connection was higher than that of the zirconia abutment with a tube in tube connection. On the other hand, the risk of long-term mechanical and biological complications is expected to decrease using zirconia abutment with an internal conical connection versus tube in tube connection. Consequently, implant platform switching, which has been proven to have biological advantages over platform matching, and helps to minimize the marginal bone loss, does not seem to be a critical parameter for the mechanical integrity of the implants. It seems that the configuration and design of the connection is more important than the diameter of the abutment. Moreover, there are no significant differences between the two all-ceramic crowns. Both monolithic zirconia and monolithic lithium disilicate have the same mechanical behavior.

Additional research is needed to confirm these results to other systems and to evaluate the differences after the simulated clinical function. It is difficult to predict the clinical results on the basis of in vitro studies, as many factors affect the oral cavity.

## 5. Conclusions

Within the limitations of this in vitro study, the following can be concluded:CAD/CAM zirconia abutments with conical implant−abutment connections for platform switching supporting single all-ceramic crowns made with a chairside workflow are highly reliable in terms of fracture resistance, both in static and in fatigue loads.The stress distribution affects the implant/restoration complex depending on the connection design. The component more susceptible to fracture is the screw. In the conical connection group, the screw is subjected to a lower stress level.The mechanical behavior of the restorative complex does not seem to be affected by the different crown materials (zirconia or lithium disilicate) used in this study.

## Figures and Tables

**Figure 1 materials-14-05009-f001:**
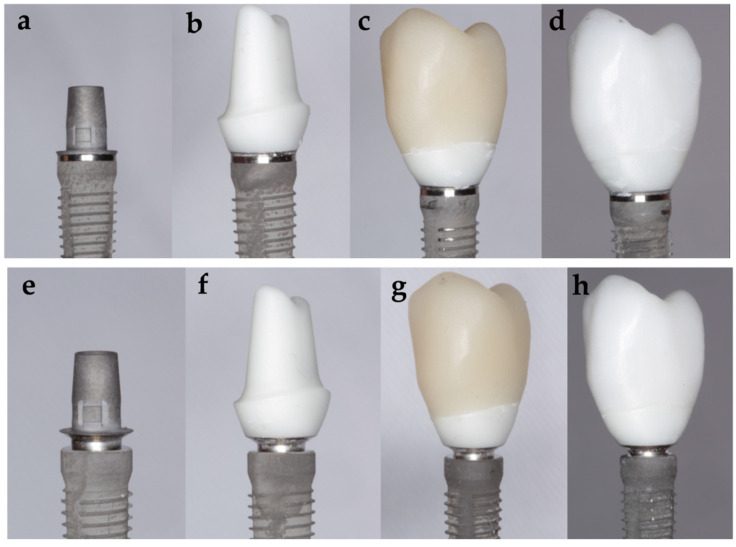
Test groups: (**a**) Tube−tube implant−abutment connection; (**b**) zirconia abutment; (**c**) lithium disilicate crown (Tube ZLD); (**d**) zirconia crown (Tube ZZ); (**e**) conical implant−abutment connection for platform switching supporting; (**f**) zirconia abutment; (**g**) lithium disilicate crown (Conical ZLD); (**h**) zirconia crown (Conical ZZ).

**Figure 2 materials-14-05009-f002:**
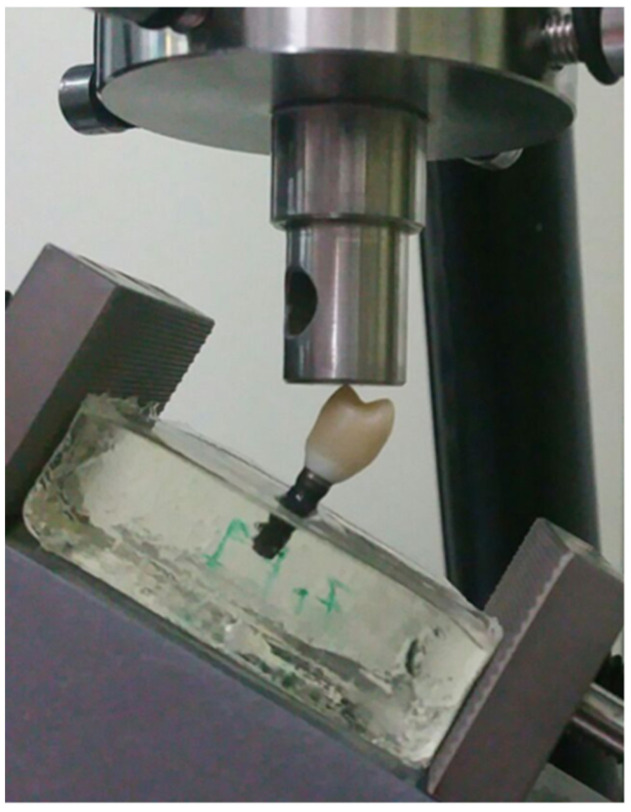
Test set-up following ISO 14801.

**Figure 3 materials-14-05009-f003:**
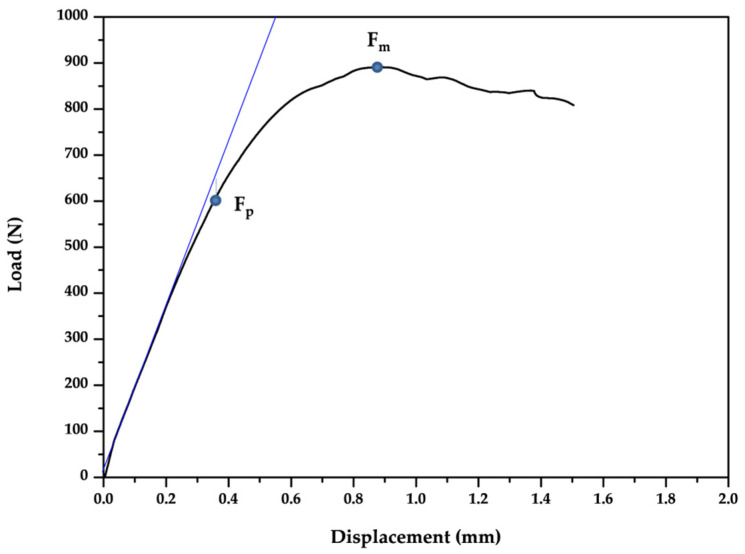
Representative load–displacement curve of a specimen. The force (F_p_) at which the load–displacement curve first deviated by 10% from the regression line in the elastic interval was recorded as an indicator for plastic deformation. The maximum force (F_m_) registered was regarded as the load–bearing capacity.

**Figure 4 materials-14-05009-f004:**
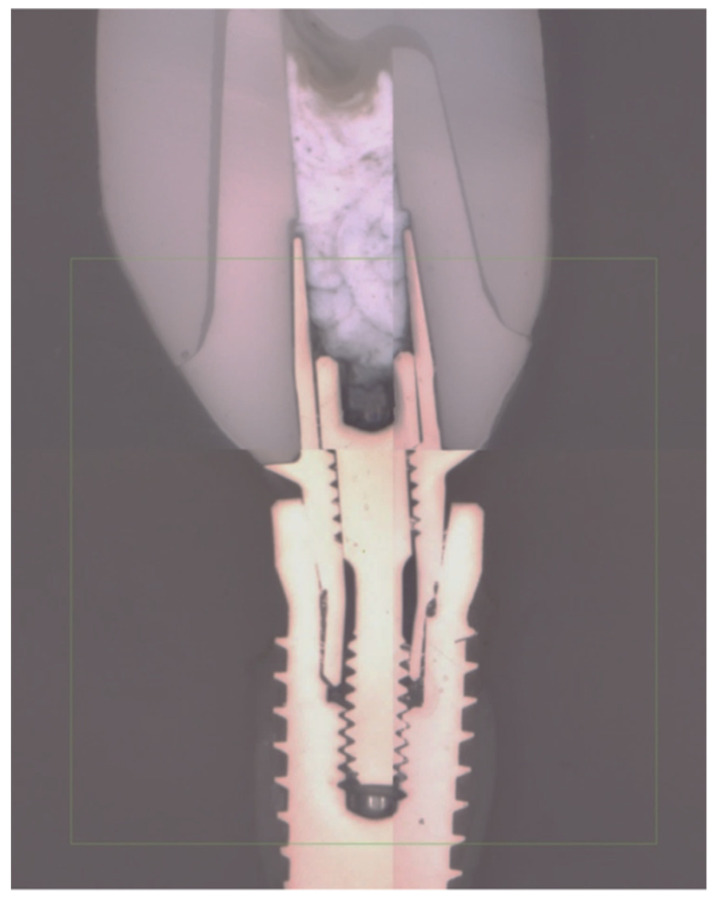
Representative cross-section micrograph of the implant–abutment–crown configuration after the static loading test.

**Figure 5 materials-14-05009-f005:**
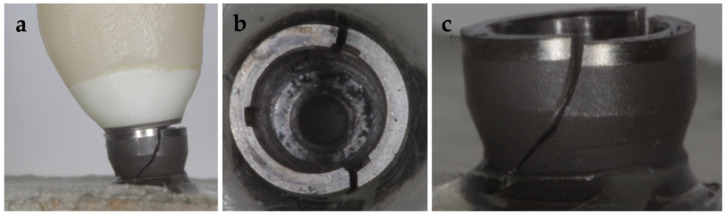
Representative image of the tube in tube internal connection specimen after the fatigue test showing fractures of the implant neck: (**a**) general view; (**b**) top view; (**c**) lateral view.

**Figure 6 materials-14-05009-f006:**
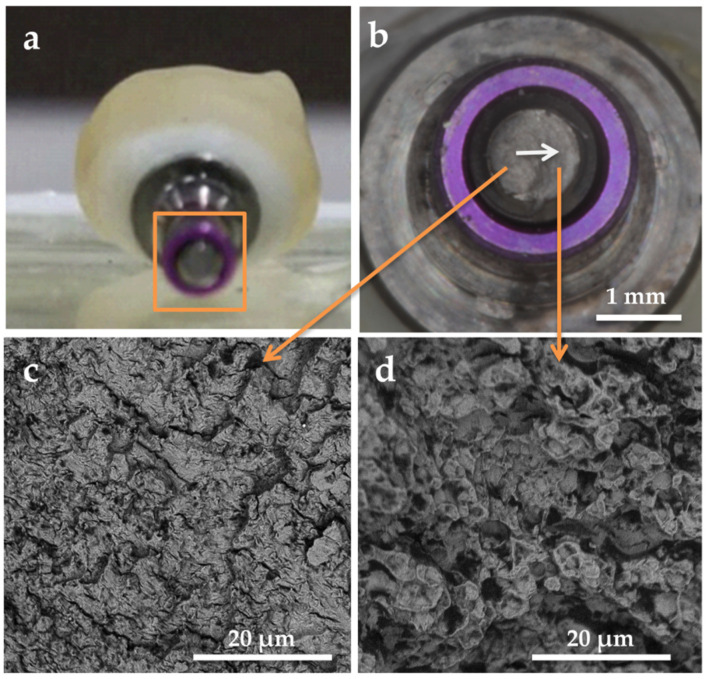
Representative image of the tube in tube internal connection specimen after the fatigue test showing an abutment screw fracture: (**a**) general view; (**b**) top view with an SEM analysis performed on each separate region of the fractured area: (**c**) fatigue striated zone and (**d**) catastrophic dimple zone. The white arrow shows the fracture direction of the screw.

**Figure 7 materials-14-05009-f007:**
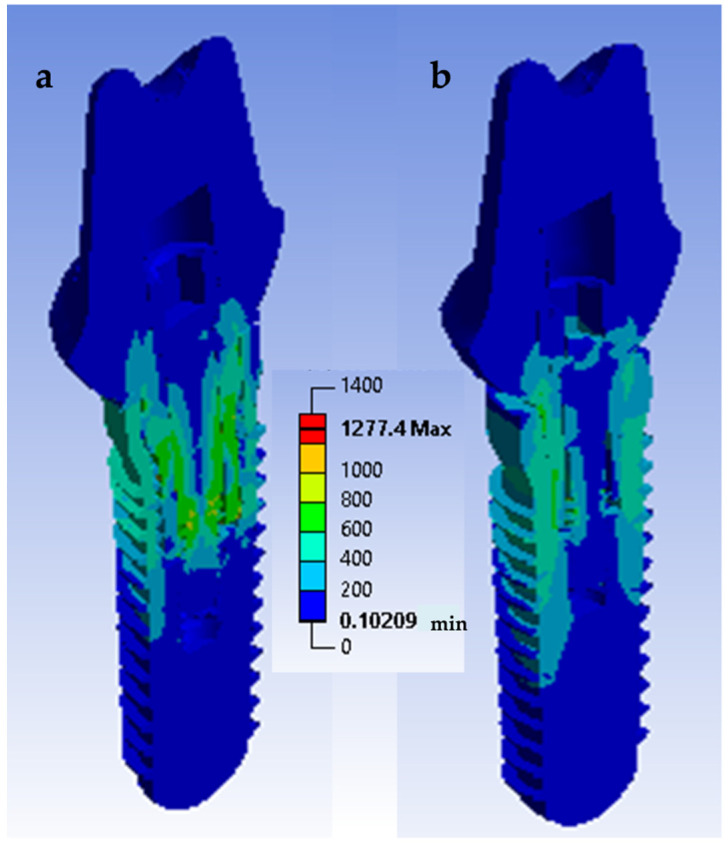
Von Mises stress distributions for (**a**) tube in tube connection and (**b**) conical connection with platform switching. A compression load of 900 N was applied to the surface of the restoration.

**Figure 8 materials-14-05009-f008:**
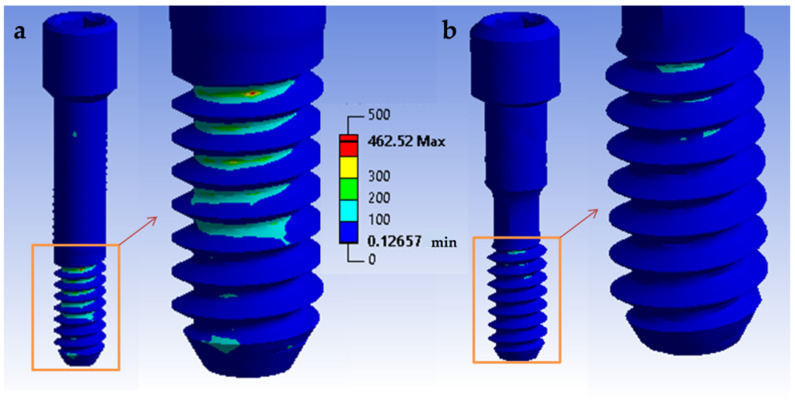
Stress distribution in abutment screw of different groups of a (**a**) tube in tube connection and (**b**) conical connection under a cyclic load of 400 N. Values are expressed by a gradient color von Mises scale.

**Figure 9 materials-14-05009-f009:**
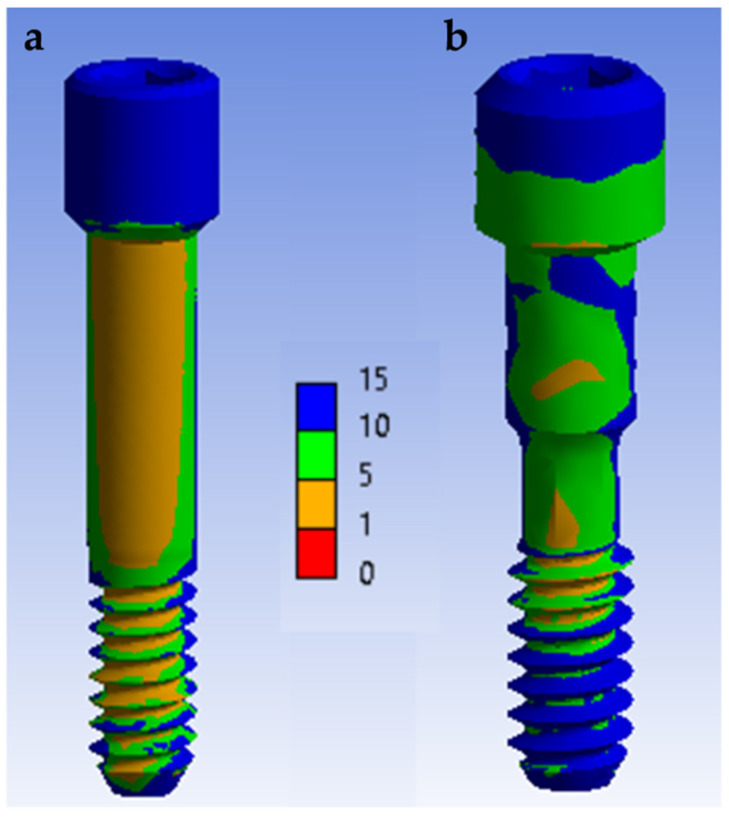
Safety factor map at a maximum cyclic load of 400 N after 2 × 10^6^ cycles: (**a**) tube in tube connection (minimum value: 0.87) and (**b**) conical connection (minimum value: 2.2). The most critical zones are indicated in orange.

**Table 1 materials-14-05009-t001:** Material properties used in the finite element models.

Material and Implant Component	Elastic Modulus (GPa)	Poisson’s Ratio	Strength (MPa)	Yield Strength (MPa)
Titanium Grade 5 (Ti base and screw) [35]	110	0.34	860	790
Titanium Grade 4 (implant body) [35]	110	0.34	550	480
Zirconia (abutment and crown) [35]	210	0.31	1000	-
Lithium disilicate (crown) *	95	0.23	360	-
Epoxy resin (specimen holder) *	4	0.33	-	-

* Values provided by the manufacturer.

**Table 2 materials-14-05009-t002:** Nodes and elements.

Model	Nodes	Elements	Minimum Edge Length (µm)
Tube connection	151,091	91,032	5.6
Conic connection	95,726	55,521	1.4

**Table 3 materials-14-05009-t003:** Mechanical properties.

Subgroup	Maximum Load (F_m_) N	Load at Plastic Deformation (F_p_) N	Survival Cycles at a Maximum Load of 400 N
Conical ZZ	1050 ± 92	704 ± 44	2 × 10^6^
Conical ZLD	1102 ± 112	682 ± 64	2 × 10^6^
Tube ZZ	1015 ± 73	552 ± 87	403.673 ± 85.398
Tube ZLD	975 ± 64	563 ± 66	254.577 ± 57.789

## Data Availability

The data presented in this study are available upon request from the corresponding author.

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
