# Peer review of "Mechanical Performance of Chairside Ceramic CAD/CAM Restorations and Zirconia Abutments with Different Internal Implant Connections: In Vitro Study and Finite Element Analysis"

_materials, 2021, doi:10.3390/ma14175009_

Round 1
Reviewer 1 Report
Care should be taken to comply with the requirements for editing the article (font size used for figures, relative to the font size of the text).
Author Response
Reviewer #1:
Care should be taken to comply with the requirements for editing the article (font size used for figures, relative to the font size of the text).
Thank you for your suggestion. We have corrected it.

Reviewer 2 Report
This study assessed the in vitro mechanical behaviour of a combination of a zirconia abutment and monolithic all-ceramic zirconia and lithium disilicate crown manufactured with a chairside workflow, connected to titanium implants with two types of internal connection. The research design is apt and the results are presented in a prominent manner. I do not have any major comments. I have a minor but important comment -
Please mention SEM micrographs which shows the crack region in the Implant neck and abutment screw. It would be interesting to observe the nature of crack propagation and material deformation.
Author Response
Reviewer #2:
This study assessed the in vitro mechanical behaviour of a combination of a zirconia abutment and monolithic all-ceramic zirconia and lithium disilicate crown manufactured with a chairside workflow, connected to titanium implants with two types of internal connection. The research design is apt and the results are presented in a prominent manner. I do not have any major comments.
Thank you very much for the positive impression of the reviewer of our paper.
I have a minor but important comment -
Please mention SEM micrographs which show the crack region in the Implant neck and abutment screw. It would be interesting to observe the nature of crack propagation and material deformation.
Thank you for the remark. We have divided the crack region of the implant neck and abutment screw into two independent figures to improve clarity. Moreover, we have marked the fractured area in the SEM micrographs as suggested.

Reviewer 3 Report
In the present work the authors investigated the mechanical properties and behaviour of a combination of a zirconia abutment and monolithic all-ceramic zirconia and lithium disilicate crown, both with two types of internal connection (tube in tube and conical connections). They performed static and fatigue mechanical
tests, microstructural analysis using a scanning electron microscope (SEM) and finite element simulations to obtain the stress distribution.
The work is quite interesting with a comparison of different techniques; however in my opinion some additional efforts are needed to improve the study and report complete and significant outcomes:
- the authors are “assessing the mechanical behaviour of the materials” but no stress/strain or force/displacement curves are reported. Since they referred to some results, I believe they should add also this information (curves of both static and fatigue tests), with also the regressions they used to obtain the results and a table with the mechanical properties. Are they the same used in the FEM?
- Referring to FEA, I did not understand why they refer to numerical simulations as “qualitative” results. In my opinion they should provide quantitative results also, after a good problem description and parameters calibration.
- The authors said they performed a microstructural analysis with SEM, but the results are not clear to me, especially Fig. 4B. I suggest the authors to add a descriptive part about it and perhaps more pictures.
- I would spend few words more in adding info about the FEM descriptions (type of analysis, time steps, load, element types etc..)
- I think the aims part in the introduction should be improved.
- Some typos to correct e.g., line 50, line 77, 164, 261 etc..
Author Response
Reviewer #3:
In the present work the authors investigated the mechanical properties and behaviour of a combination of a zirconia abutment and monolithic all-ceramic zirconia and lithium disilicate crown, both with two types of internal connection (tube in tube and conical connections). They performed static and fatigue mechanical tests, microstructural analysis using a scanning electron microscope (SEM) and finite element simulations to obtain the stress distribution.
The work is quite interesting with a comparison of different techniques; however in my opinion some additional efforts are needed to improve the study and report complete and significant outcomes:
- The authors are “assessing the mechanical behaviour of the materials” but no stress/strain or force/displacement curves are reported. Since they referred to some results, I believe they should add also this information (curves of both static and fatigue tests), with also the regressions they used to obtain the results and a table with the mechanical properties. Are they the same used in the FEM?
Thank you for the observation. We have added a new figure with a representative load/displacement curve (Figure 3).
Revised text:
Figure 3 Representative load–displacement curve of a specimen. The force, Fp, at which the load–displacement curve first deviated by 10% from the regression line in the elastic interval was recorded as indicator for plastic deformation. The maximum force (Fm) registered was regarded as load–bearing capacity.
Additionally, we have added a new table (Table 3) with all the obtained mechanical properties as suggested.
Revised text:
Table 3. Mechanical properties.
|
Subgroup |
Maximum load (Fm) N |
Load at plastic deformation (Fp) N |
Survival cycles at a maximum load of 400 N |
||
|
Conical ZZ |
1050±92 |
704± 44 |
2·106 |
||
|
Conical ZLD |
1102±112 |
682± 64 |
2·106 |
||
|
Tube ZZ |
1015±73 |
552± 87 |
403.673±85.398 |
||
|
Tube ZLD |
975±64 |
563± 66 |
254.577±57.789 |
||
The mechanical properties used in finite element models (Table 1) correspond to individual components fabricated with different materials. This information has been shown in the Table. The values used have been taken from the literature (reference [35]) and provided by the manufacturers (Ivoclar Vivadent and Buehler) which have been also referenced in the Table.
Revised text:
Table 1. Material properties used in finite element models.
|
Material and implant component |
Elastic Modulus (GPa) |
Poisson´s ratio |
Strength (MPa) |
Yield strength (MPa) |
|
Titanium Grade 5 (Ti-base and screw) [35] |
110 |
0.34 |
860 |
790 |
|
Titanium Grade 4 (Implant body) [35] |
110 |
0.34 |
550 |
480 |
|
Zirconia (Abutment and crown) [35] |
210 |
0.31 |
1000 |
- |
|
Lithium disilicate (Crown)* |
95 |
0.23 |
360 |
- |
|
Epoxi Resin (Specimen holder)* |
4 |
0.33 |
- |
- |
* Values provided by the manufacturer
[35] Gultekin A, Turkoglu P, Yalcin S. Application of Finite Element Analysis in Implant Dentistry, In book: Finite Element Analysis - New Trends and Developments; Ebrahimi F. Ed.; Publisher: IntechOpen, London, UK, 2012, pp. 21-54.
- Referring to FEA, I did not understand why they refer to numerical simulations as “qualitative” results. In my opinion they should provide quantitative results also, after a good problem description and parameters calibration.
The reviewer is totally right, we have corrected it in the text.
Revised text:
Pag. 9. “A quantitative view allows for perceiving an increase in stress in tube connection subgroups when compared to conical connection subgroups.”
- The authors said they performed a microstructural analysis with SEM, but the results are not clear to me, especially Fig. 4B. I suggest the authors to add a descriptive part about it and perhaps more pictures.
Thank you for the remark. We have added a new picture (Figure 5) with a descriptive part about the microstructural SEM analysis performed on each separate region of the screw fracture surface as suggested.
- I would spend few words more in adding info about the FEM descriptions (type of analysis, time steps, load, element types etc..)
We have added more information about the FEM description.
Revised text:
Pag. 6. “It was intended to obtain the values in terms of stress for each of the specimens, and compare them with each other in static and cyclic fatigue conditions according to the ISO 1480.”
Page 6. “The element is defined by 10 nodes having three degrees of freedom at each node: translations in the nodal x, y, and z directions. The element has plasticity, hyperelasticity, creep, stress stiffening, large deflection, and large strain capabilities. It also has mixed formulation capability for simulating deformations of nearly incompressible elastoplastic materials, and fully incompressible hyperelastic materials.”
Page 9. “A quantitative view allows for perceiving an increase in stress in tube connection subgroups when compared to conical connection subgroups when a static load of 900 N is applied (Figure 7).”
Page 10. “At a cyclic load of 400N the screw suffers a maximum tensile stress of 146MPa in the conical connection model and 462MPa in the tube connection model.”
- I think the aims part in the introduction should be improved.
The aims part in the introduction has been improved.
Revised text:
“Therefore, zirconia abutments with a titanium base are promising candidates to substitute for titanium abutments from an aesthetic and biological point of view. However, information on the mechanical performance of zirconia abutments supporting ceramic crowns and the influence of the implant connection type is limited. Due to this current scarcity of mechanical in-vitro studies and lack of relevant clinical studies, fracture resistance of chairside CAD-CAM zirconia implant abutments supporting translucent zirconia and lithium disilicate monolithic crowns should be analyzed. The aim of this study is to assess the mechanical behavior of chairside customized zirconia abutments with two different implant connections under static and fatigue conditions. Moreover, the in vitro tests were reproduced using engineering tools like FEM and von Mises analysis as they can consider all the different variables (material characteristics and type of load) and predict the stress distribution of a system in which dental implant, bone tissue, and prosthodontic components are involved.”
- Some typos to correct e.g., line 50, line 77, 164, 261 etc
All the typos have been corrected in the text.

Round 2
Reviewer 3 Report
I appreciated the authors’ effort in
improving the quality of the work. I agree with the
publication of this study.
This manuscript is a resubmission of an earlier submission. The following is a list of the peer review reports and author responses from that submission.